# Causal Discovery under Changing Mechanisms: A Unified Graphical Approach

## Abstract

A common assumption of general causal mechanisms is invariant across all situations in the environment, just as Newton's laws of motion are always-valid mechanisms. However, in reality, the causal mechanisms are often partially activated from a partially observed mechanism under specific situations, such as the power mechanism of a hybrid vehicle changes according to the type of energy source available. This brings three following problems: (i) definitely, how to describe these changing mechanisms in a unified causal model, (ii) theoretically, what conditions make the dynamic causal model identifiable, and (iii) methodology, how to learn the model. In response to them, we novelly extend the definition of the directed acyclic graph to the **dynamic causal graph** with condition labels on edges. We provide the identification when the changing mechanisms follow a **linear latent Gaussian dynamic causal model (DynaCM)**. Building upon these, we devise a five-step algorithm to recover causal mechanisms and reduce condition labels of edges, thereby identifying the dynamic causal graph. Experiments on both synthetic and real-world data demonstrate the effectiveness of our method.

## 1 Introduction and Related Work

Causality seeks to uncover mechanisms that generate data, offering a structured account of how variables influence one another beyond correlations (Pearl, 2009b; Schölkopf et al., 2021). Yet in many systems, these mechanisms are not fixed but depend on conditions (Rubenstein et al., 2017; Peters et al., 2017). A parallel can be found in physics: macroscopic laws like thermodynamics appear universal, but at the microscopic level they arise from interactions that vary with states like temperature or pressure (Chalupka et al., 2016; Beckers & Halpern, 2019). If one insists on describing both levels with a single fixed law, crucial conditional details are lost. The same problem arises in causal mechanisms: observations collected in different contexts or at different scales may yield seemingly inconsistent causal graphs (Rubenstein et al., 2017). This issue is particularly prevalent in the era of large-model and complex agents decision-making scenarios, where an agent collects data in an open-world environment under a limited observation window. In such cases, the causal mechanisms themselves may shift as the observation window changes, leading to context-dependent and seemingly inconsistent causal structures.

Classical discovery methods, which assume a fixed causal structure, either merge data from different conditions, obscuring the true mechanisms, or analyze them separately, fragmenting the system. Existing approaches fall into three categories: (i) constraint-based methods (Spirtes et al., 2000; 1995; Huang et al., 2022; Dong et al., 2023), (ii) score-based methods (Alonso-Barba et al., 2013; Karan & Zola, 2016), and (iii) functional-based methods (Yang et al., 2021; Zhang & Hyvärinen, 2009). More detailed discussion of the related work is provided in Appendix A. While these methods recover causal structures under their respective assumptions, they still rely on fixed mechanisms and cannot capture condition-dependent variations. This calls for a unified framework to represent such relations within a single graph.

Take Figure 1 as an illustrated example. (i) In the condition that Switch 1 is on, Light is caused by Electricity and Switch 1. (ii) In the condition that Switch 1 is OFF and Switch 2 is ON, Light is caused by Switch 2 and Electricity. (iii) In the condition that both Switch 1 and Switch 2 are OFF, no causality exists. These changing mechanisms bring the following three problems:

1. *How to formulate the changing causal mechanisms and unknown corresponding conditions in a unified graph model?*
2. *In what situations will the dynamic graph model be identifiable?*
3. *How to identify the dynamic graph model?*

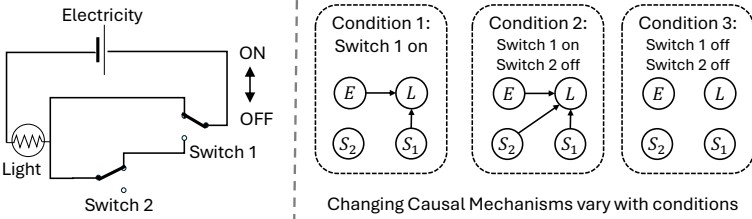

Figure 1: A simple case (from Pearl (2009b)) of dynamic mechanisms in a circuit environment and its three causal graphs under three conditions. Here, $E$ denotes Electricity, $L$ denotes Light, $S_1$ denotes Switch 1, and $S_2$ denotes Switch 2.

To address the first problem, we introduce the concept of a dynamic causal graph, which generalizes the classical invariant causal graph by allowing causal edges to change under latent condition triggers. Unlike standard Directed Acyclic Graphs (DAGs) that assume fixed mechanisms across all contexts, the dynamic causal graph explicitly represents mechanism variations induced by latent conditions. To solve the second problem, we theoretically analyze the identification of a dynamic causal graph under a linear latent dynamic Gaussian causal model assumption. Building upon this, we solve the third problem with a five-step dynamic causal discovery algorithm. We summarize our contributions as follows:

- Conceptually, we first propose the definition of a dynamic causal graph, which describes the changing mechanisms of reality in a general unified graph model.
- Theoretically, we proposed the identification of our dynamic causal discovery algorithm under a very general assumption.
- Methodologically, we proposed a five-step algorithm for learning the dynamic causal graph even if existing latent variables.

## 2 LINEAR LATENT GAUSSIAN DYNAMIC CAUSAL MODEL

In this work, we focus on the linear latent Gaussian dynamic causal model (DynaGCM), in which the changing mechanisms of the causal graph are triggered by latent conditions. We begin by defining a dynamic causal graph—capturing the causal graph skeleton of DynaGCM—based on the concept of a DAG with graph labeling (Gallian, 2012).

**Definition 1 (Dynamic causal graph)** *A dynamic causal graph $\mathcal{G}$ is a directed acyclic graph with labeled directed edges. Formally, it is a 3-tuple $\mathcal{G} = (\mathbf{v}, \mathbf{e}, \mathbb{C}^{\mathbf{e}})$ where*

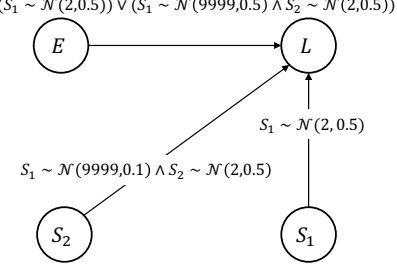

Figure 2: An example of a dynamic causal graph in a circuit environment, each of variables is described by Gaussian distribution.

- $\mathbf{v}$ *is a list of variables in the system;*
- $\mathbf{e}$ *is a list of directed edges spanning all conditions, e.g. $e_{i \rightarrow j} \in \mathbf{e}$ means an edge from $\mathbf{v}_i$ to $\mathbf{v}_j$;*
- $\mathbb{C}^{\mathbf{e}}$ *is the condition set of the existence of directed edges $\mathbf{e}$ among the distribution of data variables, such as the condition of existance of $e_{i \rightarrow k}$ is $(\mathbf{v}_i, \mathbf{v}_j, \mathbf{v}_k) \in \mathbf{v}, \mathbf{c}^{e_{i \rightarrow k}} : (\mathbf{v}_i \sim \mathcal{N}(1,3) \wedge \mathbf{v}_j \sim \mathcal{N}(5,2)) \in \mathbb{C}^{\mathbf{e}};$*

Figure 2 gives an example of a dynamic causal graph. Note that an 9999 means that the switch is open. Let $\mathbf{o}$ and $\mathbf{l}$ represent the observed and latent variables, respectively. Then, their concatenation $\mathbf{v} := \mathbf{o} \frown \mathbf{l}$ represents the total variables in $\mathcal{G}$. Let $\mathbf{x} := \mathbf{x}^{\mathbf{o}} \frown \mathbf{x}^{\mathbf{l}}$ be the dataset, where $\mathbf{x}^{\mathbf{o}}$ and $\mathbf{x}^{\mathbf{l}}$ are the samples on the observed variables and latent variables, respectively. Then the linear latent Gaussian dynamic causal model can be defined as follows:

**Definition 2 (Linear Latent Gaussian Dynamic Causal Model (DynaGCM))** *Suppose a dynamic causal graph $\mathcal{G} = (\mathbf{v}, \mathbf{e}, \mathbb{C}^{\mathbf{e}})$, each data variable is generated by the equation:*

$$\mathbf{v}_i = \sum_{\mathbf{v}_j \in Pa(\mathbf{v}_i), f_{ver}(\mathbf{x}^{\mathbf{o}}, \mathbf{c}^{\mathbf{e}_{j \to i}}) = 1} \alpha^{\mathbf{e}_{j \to i}} \mathbf{v}_j + \epsilon_i, \tag{1}$$

*where $Pa(\mathbf{v}_i) = \{\mathbf{v}_j \mid (\mathbf{v}_j \to \mathbf{v}_i) \in \mathbf{e}, 0 < \Pr_{\mathbf{x}^{\mathbf{o}}}[f_{ver}(\mathbf{x}^{\mathbf{o}}, \mathbf{c}^{\mathbf{e}_{j \to i}}) = 1]\}$ denotes the set of all the parents nodes of $\mathbf{v}_i$ regardless condition, and $\alpha^{\mathbf{e}_{j \to i}} \in [0, 1]$ is the linear weight of edge $\mathbf{e}_{j \to i}$. $f_{ver}(\mathbf{x}^{\mathbf{o}}, \mathbf{c}^{\mathbf{e}_{j \to i}})$ indicates whether the observed value of variables in sample $\mathbf{x}$ satisfy the condition $c^{\mathbf{e}_{j \to i}}$. It is defined as*

$$f_{ver}(\mathbf{x}^{\mathbf{o}}, \mathbf{c}^{\mathbf{e}_{j \to i}}) = \begin{cases} 1, & \textit{if the observed variables in sample } \mathbf{x}^{\mathbf{o}} \textit{ satisfy } c^{\mathbf{e}_{j \to i}}, \\ 0, & \textit{otherwise.} \end{cases} \tag{2}$$

*Moreover, each noise variable $\epsilon$ is a Gaussian distribution variable.*

As shown in Figure 1, if one turns on Switch 1, it changes the state of Switch 1 and the existence of the causal edge between Switch 2 and Light; we call this edge as Dynamic Edge defined as below.

**Definition 3 (Dynamic Edge)** *Given a $\mathcal{G} = (\mathbf{v}, \mathbf{e}, \mathbb{C}^{\mathbf{e}})$. An edge $\mathbf{e} \in \mathbf{e}$ is called a dynamic edge iff the activation is not unique across samples, i.e., $0 < \Pr_{\mathbf{x}^{\mathbf{o}}}[f_{ver}(\mathbf{x}^{\mathbf{o}}, \mathbf{c}^{\mathbf{e}}) = 1] < 1$.*

For constructing DynaGCM, the main challenge lies in identifying dynamic edges and their associated conditions. Essentially, this amounts to uncovering the relevant internal states: the activation of a dynamic edge is triggered by changes in these internal states, which can be traced back to the root causes of the variables influencing the edge and their corresponding noise terms. Consequently, we determine the conditions of dynamic edges (in the following definition) through specific observation values that serve as proxies for these internal states.

**Definition 4 (The Condition of a Dynamic Edge)** *The condition of a dynamic edge is a minimal set of multivariable Gaussian distributions on some observed variables, such that:*

*(i) If these multivariable Gaussian distributions hold in the environment, the dynamic edge exists. If the dynamic edge does not exist, these multivariable Gaussian distributions must not hold.*

*(ii) The changes in the observed Gaussian distributions are caused by changes in their noise variables rather than the influences from other variables.*

The first rule ensures that one can infer if a dynamic edge is activated from his observed observation. The second rule ensures that inference is based on the root cause that activates the dynamic edge. Building upon the above definitions, we conclude the problem of dynamic causal discovery as:

**Definition 5 (The Problem of Linear Latent Dynamic Causal Discovery)** *Given the observed data of variables $\mathbf{x}^{\mathbf{o}}$ which is generated under Equation 1, the problem of dynamic causal discovery is to identify the tuple $\mathcal{G} = (\mathbf{v}, \mathbf{e}, \mathbb{C}^{\mathbf{e}})$.*

## 3 PRELIMINARY INTUITION AND PROBLEM DECOMPOSITION OF DISCOVER DYNAGCM

In this section, we present the general idea for solving the DynaGCM discovery problem (Definition 5) to help understand what we do. The key is to construct the dynamic causal graph from multiple linear latent Gaussian causal models (GCMs) (Dong et al., 2025), which serve as the static counterparts of our proposed DynaGCM. Take Figure 1 as an example. In the circuit environment, there are three possible causal relationships. Each sample $\mathbf{x}$ follows one of the three conditions in Figure 1, and its generation equation (Eq. 1) can be written as

$$\mathbf{v}_i = \sum_{\mathbf{v}_j \in Pa(\mathbf{v}_i), f_{ver}(\mathbf{x}, \mathbf{v}_{s1}=1)=1} \alpha^{\mathbf{e}_{j \to i}} \mathbf{v}_j + \epsilon_i^1, \tag{3}$$

$$\mathbf{v}_i = \sum_{\mathbf{v}_j \in Pa(\mathbf{v}_i), f_{ver}(\mathbf{x}, \mathbf{v}_{s1}=1 \wedge \mathbf{v}_{s2}=0)=1} \alpha^{\mathbf{e}_{j \to i}} \mathbf{v}_j + \epsilon_i^2, \tag{4}$$

or

$$\mathbf{v}_i = \sum_{\mathbf{v}_j \in Pa(\mathbf{v}_i), f_{ver}(\mathbf{x}, \mathbf{v}_{s1}=0 \wedge \mathbf{v}_{s2}=0)=1} \alpha^{\mathbf{e}_{j \to i}} \mathbf{v}_j + \epsilon_i^3, \tag{5}$$

where the index $\{1, 2, 3\}$ corresponds to Condition 1, 2, or 3, respectively. Each equation with its DAG defines a GCM, which can be recovered by the RLCD algorithm (Dong et al., 2023) and the parameter estimation method (Dong et al., 2025). In other words, if the assignment of each data point to its condition group is known, existing methods already provide an identification scheme.

Given the identification of GCM, our approach proceeds in three steps: (i) assign the observed data into groups for distinct conditions, (ii) discover a GCM within each group, and (iii) merge the edge conditions across groups to define the dynamic edge condition. This raises two key problems: (i) what is the identification condition for assigning samples to GCMs, and how to recover these assignments, and (ii) how to reduce multiple GCMs into a dynamic causal graph.

For (i), the identification theory in Dong et al. (2025) shows that different GCMs lead to different Gaussian distributions of $\mathbf{x^o}$, while Yakowitz & Spragins (1968) establishes identifiability for mixtures of multivariable Gaussian distributions. Together, they justify using mixture Gaussian learning methods, such as the Expectation-Maximization (EM) algorithm, to recover the assignments. For (ii), inspired by Definition 4, we infer the condition sets $\mathbb{C}^\mathbf{e}$ of a dynamic causal graph $\mathcal{G}$ from differences among estimated GCMs: if a minimal set of noise distributions determines the existence of an edge, then the observed distributions form a condition $\mathbf{c^e} \in \mathbb{C}^\mathbf{e}$ for that edge. Detailed explanation please check Appendix E. We provide a detailed related work discussion with us in Appendix A

## 4   DYNAMIC CAUSAL DISCOVERY ALGORITHM

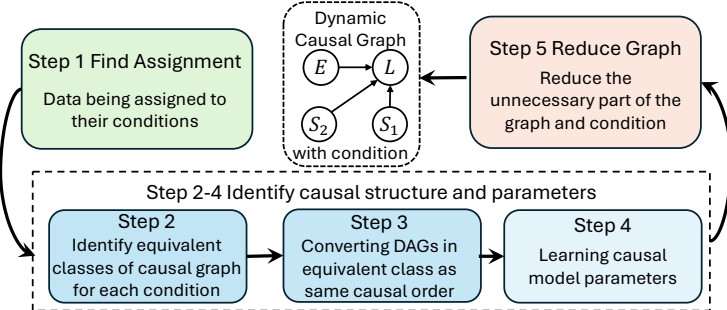

Figure 3: Flowchart of Our Algorithm.

The above discussion given in Section 2 and 3 inspires our solution to the linear latent dynamic causal discovery problem. We provided a flowchart of our algorithm in Figure 3, which can be summarized as five steps: (i) learning sample assignments of GCMs ("*find Assignments*"), assign the observed data into groups corresponding to distinct conditions (ii) identify Markov equivalent classes of GCMs for each assignment ("*identify Causal Structures*"), (iii) converting each Markov equivalent class to DAG in a same causal order ("*PDAGs To DAGs*"), (iv) learning parameters of GCMs with given DAGs ("*identify Causal Models*"), (v) reducing the dynamic causal graph and conditions, i.e., DynaGCM ("*reduce Dynamic Causal Graph*"). We propose our method in Algorithm 1 and detail each step in the following subsections.

### 4.1   STEP 1: FIND ASSIGNMENTS

We start by learning the assignments of observed data $\mathbf{x^o}$ in this step. Given that $\mathbf{x^o}$ is generated by an DynaCM with dynamic causal graph $\mathcal{G}$, we assume there are $K$ different GCMs in the environment, and thereby $\mathbf{x^o}$ follows a mixture Gaussian distribution with $K$ components.

To identify the assignments of each sample, we begin with the identification of the mixture Gaussian distribution. Teicher (1961) proves that a one-dimensional mixture Gaussian distribution is identifiable up to label permutation. Furthermore, Yakowitz & Spragins (1968) relaxes the identifiable theory to the multi-dimensional setting, which can be described as follows:

**Theorem 1 (Identifiability of Multivariate Gaussian Mixtures (Yakowitz & Spragins, 1968))**
*Let $\mathcal{N}_d = \{\mathcal{N}(\cdot \mid \mu, \Sigma) : \mu \in \mathbb{R}^d, \Sigma \succ 0\}$ be a set of linear-independence $d$-dimensional Gaussian distribution. Finite mixtures over $\mathcal{N}_d$ are identifiable up to label permutation; i.e., if two finite Gaussian mixtures induce the same distribution, then after relabeling they have identical numbers of components and matching mixing weights, means, and covariance matrices.*

---

**Algorithm 1:** Dynamic Causal Discovery

---

**Require:** $N$ observed samples $\mathbf{x^o} = \{x_n^o\}_{n=1}^N$ on variables $\mathbf{o}$
**Ensure:** dynamic causal graph $\mathcal{G}$
1: //— $K$ = #GCMs, $h_n \in \mathbf{h}$ is assignment of $x_n^o$
2: $(K, \mathbf{h}) \leftarrow$ FINDASSIGNMENTS($\mathbf{x^o}$)
3: **for each** $k$ **in** $\{1:K\}$ **do**
4:   $\hat{\mathbf{x^o}} \leftarrow \{x_n^o \mid h_n = k, 1 \le n \le N\}$
5:   $\mathcal{M}_k \leftarrow$ IDENTIFYCAUSALSTRUCTURES($\hat{\mathbf{x^o}}$)
6: **end for**
7: $\mathcal{D}_{1:K} \leftarrow$ PDAGSTODAGS($\mathcal{M}_{1:K}$)
8: **for each** $k$ **in** $\{1:K\}$ **do**
9:   $\hat{\mathbf{x^o}} \leftarrow \{x_n^o \mid h_n = k, 1 \le n \le N\}$
10:   //— parameters $\Theta_k$ for $k^{\text{th}}$ model
11:   $\Theta_k \leftarrow$ IDENTIFYCAUSALMODELS($\hat{\mathbf{x^o}}, \mathcal{D}_k$)
12: **end for**
13: $\mathcal{G} \leftarrow$ REDUCEDYNAMICCAUSALGRAPH($\mathcal{M}_{1:K}, \Theta_{1:K}$)
14: **return** $\mathcal{G}$

---

Theorem 1 guarantees that the assignments are identifiable if the observed Gaussian distributions corresponding to different GCMs are linearly independent. This leads to the central question: under what conditions can each individual GCM be identified from the observed data?

To address this question, we introduce the concept of *atomic cover* (Dong et al., 2025). Let the *effective cardinality* be defined as $||\mathbb{V}|| = |(\cup_{\mathbf{v} \in \mathbb{V}} \mathbf{v})|$. Pure children ($PCh$) is defined as follows:

**Definition 6 (Pure Children (Dong et al., 2023))** $\tilde{\mathbf{v}}$ *are pure children of variables* $\mathbf{v}$ *in graph* $\mathcal{D}$, *iff* $Pa(\tilde{\mathbf{v}}) = \cup_{\tilde{v}_i \in \tilde{\mathbf{v}}} Pa(\tilde{v}_i) = \mathbf{v}$ *and* $\mathbf{v} \cap \tilde{\mathbf{v}} = \emptyset$. *We denote the pure children of* $\mathbf{v}$ *in* $\mathcal{D}$ *by* $PCh(\mathbf{v})$.

**Definition 7 (Atomic Cover (Dong et al., 2023))** *Let* $\hat{\mathbf{v}} \in \mathbf{v}$ *denotes the variables, where* $n_l$ *out of* $\hat{\mathbf{v}}$ *are latent, and the remaining* $|\hat{\mathbf{v}}| - n_l$ *are observed.* $\hat{\mathbf{v}}$ *is an atomic cover if* $\hat{\mathbf{v}}$ *is a single observed variable, or if the following conditions hold:*

*(i) There exists a set of atomic covers* $\mathbb{A}$, *with* $||\mathbb{A}|| \ge n_l + 1$, *such that* $\cup_{\mathbf{a} \in \mathbb{A}} \mathbf{a} \subseteq PCh(\hat{\mathbf{v}})$.

*(ii) There exists a set of covers* $\mathbb{B}$ *with* $||\mathbb{B}|| \ge n_l + 1$, *s.t.* $(\cup_{\mathbf{b} \in \mathbb{B}} \mathbf{b}) \cap (\cup_{\mathbf{a} \in \mathbb{A}} \mathbf{a}) = \emptyset$, *every element in* $\cup_{\mathbf{b} \in \mathbb{B}} \mathbf{b}$ *is a neighbour of every element in* $\hat{\mathbf{v}}$, *and* $\mathbf{v}$ *d-separates* $\mathbb{B}$ *and* $\mathbb{A}$.

*(iii) There does not exist a partition* $\mathbb{P}$ *of* $\mathbf{v}$, *s.t., all elements in* $\mathbb{P}$ *are atomic covers.*

Based on atomic cover, Dong et al. (2023) first proposes two conditions for the causal structure identification of GCM as follows:

**Condition 1 (Basic Conditions for Structure Identifiability (Dong et al., 2023))** *A causal graph* $\mathcal{D}$ *of GCM satisfies the basic graphical condition for identifiability, if every latent variable belongs to at least one atomic cover in* $\mathcal{D}$ *and for each atomic cover with latent variables, any of its children is not adjacent to any of its neighbours.*

**Condition 2 (Condition on Colliders (Dong et al., 2023))** *In* $\mathcal{D}$, *if (i) there exists sets of variables* $\mathbf{v}$, $\mathbf{v}_1$, $\mathbf{v}_2$, *and* $\mathbf{t}$ *such that every variable in* $\mathbf{v}$ *is a collider of two atomic covers* $\mathbf{v}_1$, $\mathbf{v}_2$, *and* $\mathbf{t}$ *is a minimal set of variables that d-separates* $\mathbf{v}_1$ *from* $\mathbf{v}_2$, *and (ii) there exists at least one latent variable in* $\mathbf{v} \cup \mathbf{v}_1 \cup \mathbf{v}_2 \cup \mathbf{t}$, *then we must have* $|\mathbf{v}| + |\mathbf{t}| \ge |\mathbf{v}_1| + |\mathbf{v}_2|$.

The structure identifiability result can be summarized as follows. For a DAG $\mathcal{D}$, if Condition 1 and Condition 2 hold, then the structure is asymptotically identifiable up to the Markov equivalence class (MEC) of $\mathcal{O}_{\min}(\mathcal{O}_s(\mathcal{D}))$ (see Appendix B for definitions of $\mathcal{O}_{\min}(\cdot)$ and $\mathcal{O}_s(\cdot)$). In other words, the causal structure of $\mathcal{D}$ can be recovered except that the orientation of some edges may remain undetermined. Furthermore, Dong et al. (2025) show that, for any DAG in the identified equivalence class, the parameters and noise distributions of a GCM are also identifiable from the observed data, provided that certain trek-related conditions (Sullivant et al., 2010) are satisfied. This result is formalized in Theorem 2.

**Definition 8 (Treks (Sullivant et al., 2010))** *In $\mathcal{D}$, a trek from $\mathrm{v}_i$ to $\mathrm{v}_j$ is an ordered pair of directed paths $(\mathrm{p}_i, \mathrm{p}_j)$ where $\mathrm{p}_i$ has a sink $\mathrm{v}_i$, $\mathrm{p}_j$ has a sink $\mathrm{v}_j$, and $\mathrm{p}_i$ and $\mathrm{p}_j$ have the same source $\mathrm{v}_k$, i.e., $top(\mathrm{p}_i, \mathrm{p}_j) = \mathrm{v}_k$. A Trek is simple if $\mathrm{p}_i$ and $\mathrm{p}_j$ have no intersection except their common source $\mathrm{v}_k$.*

**Theorem 2 (Sufficient Condition for Parameter Identifiability (Dong et al., 2025))** *Assume that $\mathcal{D}$ satisfies Conditions 1 and 2 and thus the structure can be asymptotically identified up to the MEC of $\mathcal{O}_{min}(\mathcal{O}_s(\mathcal{D}))$. For any DAG in the equivalence class, the parameters are identifiable (up to group sign), if both the following hold:*

*(i) For any atomic cover $\mathbf{v} = \mathbf{o} \cup \mathbf{l}$, $|\mathbf{l}| \leq 1$.*

*(ii) If an atomic cover $\mathbf{v} = \mathbf{o} \cup \mathbf{l}$ satisfies $|\mathbf{l}| \neq 0$ and $|\mathbf{o}| \geq 1$, then all simple treks between $\mathbf{l}$ and $\mathbf{o}$ do not contain any latent variables that are not in $\mathbf{l}$.*

With Theorem 1 and 2 together, we propose the identification of the observed mixture Gaussian distribution generated by DynaCM as below:

**Theorem 3 (Identification of Observed Mixture Gaussian Distribution)** *Given the observed data generated by an DynaCM, if each decomposed GCM follows the identification in Theorem 2, then the observed mixture Gaussian distribution is identifiable.*

Based on this theorem, we propose the algorithm for step 1 with the EM algorithm and Bayesian Information Criterion (BIC) penalty in Algorithm 2 to find the final assignment.

---

**Algorithm 2:** FIND ASSIGNMENTS

---

**Require:** $N$ observed samples $\mathbf{x}^{\mathbf{o}} = \{\mathrm{x}_n^{\mathbf{o}}\}_{n=1}^N$ on variables $\mathbf{o}$
**Ensure:** GCMs count $K$, assignment $\mathbf{h}$ of $\mathbf{x}^{\mathbf{o}}$
 1: $k \leftarrow 1, L_0' \leftarrow -\infty$
 2: **while** True **do**
 3:    //— EM on current $k$-component mixture
 4:    run EM until $\big|L_k^{(t)} - L_k^{(t-1)}\big| / |L_k^{(t-1)}| < 10^{-6}$, get $\{\mathrm{p}_j, \mu_j, \Sigma_j\}_{j=1}^k$
 5:    append BIC penalty on $L_k$ to get $L_k'$
 6:    **if** $L_k' \leq L_{k-1}'$ **then**
 7:      //— no further gain
 8:      set $K$ as $k-1$ and $\{\mathrm{p}_j, \mu_j, \Sigma_j\}_{j=1}^K$ as the parameters under $k-1$
 9:      **break**
10:    **else**
11:      $k \leftarrow k+1$
12:    **end if**
13: **end while**
14: set $\mathrm{h}_i \in \mathbf{h}$ as the Maximum A Posteriori (MAP) assignment of $\mathrm{x}_i^{\mathbf{o}}$ under $\{\mathrm{p}_j, \mu_j, \Sigma_j\}_{j=1}^K$
15: **return** $K, \mathbf{h}$

---

## 4.2 STEP 2-4: IDENTIFY CAUSAL STRUCTURES AND PARAMETERS

Based on Theorem 2, we identify the MEC $\mathcal{M}_k \in \mathcal{M}_{1:K}$ for each GCM by RLCD method, where the RLCD algorithm is detailed in Dong et al. (2023).

Then, for each MEC $\mathcal{M}_k \in \mathcal{M}_{1:K}$ we select a DAG $\mathcal{D}_k$, as required in Theorem 2. This corresponds to converting a Partially Directed Acyclic Graph (PDAG) into a DAG, which can be solved by the Dor–Tarsi method (Dor & Tarsi, 1992). The method proceeds iteratively: (i) identify a potential sink node (a node with no outgoing edges), and (ii) orient all adjacent undirected edges toward this sink and then remove it from the graph. However, directly applying the Dor–Tarsi method is not sufficient in the dynamic causal discovery setting, since the choice of topological order affects the estimation of parameters and noise distributions. For example, the Markov equivalence class $\mathrm{v}_i - \mathrm{v}_j - \mathrm{v}_k$ can be oriented as $\mathrm{v}_i \rightarrow \mathrm{v}_j \rightarrow \mathrm{v}_k$ or $\mathrm{v}_i \leftarrow \mathrm{v}_j \leftarrow \mathrm{v}_k$, both consistent with the same independencies but implying different topological orders and noise assignments (e.g., $\mathrm{v}_i = \epsilon_i$ vs. $\mathrm{v}_i = \alpha \mathrm{v}_j + \epsilon_i'$). This will bring the changes to the estimations on both parameter and noise variable distributions when different GCMs of DynaCM are estimated under different topological orders, causing the

spurious condition reduction of a dynamic edge. Fortunately, MECs from GCMs correspond to the same dynamic causal graph, thereby they are partially activated by the same DAG, which admits a consistent topological order. Based on this, we propose a synchronized PDAG-to-DAG procedure as our **Step 3** (Algorithm 3) that first finds a shared topological order across all MECs and then applies the Dor–Tarsi method under this order.

---

**Algorithm 3:** SYNCHRONIZED PDAGs TO DAGs

---

**Require:** PDAGs $\mathcal{M}_{1:K}$ over the same $\mathbf{v}$
**Ensure:** shared topological order $\pi$ and aligned DAGs $\mathcal{D}_{1:K}$
 1: **Fix** a tie-breaking list $\sigma = (\mathsf{v}_1, \ldots, \mathsf{v}_{|\mathbf{v}|})$ covering $\mathbf{v}$ (e.g., lexicographic by variable name).
 2: $\pi \leftarrow [\,]$; make working copies $\widetilde{\mathcal{M}}_{1:K}$ on $\mathcal{M}_{1:K}$
 3: **while** $|\pi| < |\mathbf{v}|$ **do**
 4:     // Intersection of Dor–Tarsi potential sinks across all PDAGs (Dor & Tarsi, 1992)
 5:     $\mathbb{S} \leftarrow \bigcap_{i=1}^{K} \text{POTENTIALSINKS}_{\text{DT}}(\widetilde{\mathcal{M}}_i)$
 6:     **if** $\mathbb{S} = \emptyset$ **then**
 7:         **backtrack** or **fail**
 8:     **end if**
 9:     $v \leftarrow \min_{\prec_\sigma} \mathbb{S}$
10:     **for** $i = 1$ to $K$ **do**
11:         // one Dor–Tarsi elimination step at $v$ (Dor & Tarsi, 1992)
12:         $\widetilde{\mathcal{M}}_i \leftarrow \text{DORTARSISTEP}(\widetilde{\mathcal{M}}_i, v)$
13:         remove $v$ from $\widetilde{\mathcal{M}}_i$
14:     **end for**
15:     append $v$ to $\pi$
16: **end while**
17: // finalize: execute the fixed order on originals (Dor & Tarsi, 1992)
18: $\mathcal{D}_{1:K} \leftarrow \{\text{DORTARSIRUNWITHORDER}(\mathcal{M}_i, \pi)\}_{i=1}^{K}$
19: **return** $(\pi, \mathcal{D}_{1:K})$

---

The proposed PDAG-to-DAG procedure resolves the ambiguity in topological ordering and thereby avoids spurious changes in noise variables. With the identifiability guaranteed by Theorem 2, in **Step 4**, we estimate the parameters and noise distributions using stochastic gradient descent. To this end, we first reformulate a GCM, as given in Equation 3, as follows:

$$\mathbf{v} = \mathbf{F}^\top \mathbf{v} + \boldsymbol{\epsilon}, \qquad \boldsymbol{\epsilon} \sim \mathcal{N}(\boldsymbol{\mu}_\epsilon, \boldsymbol{\Sigma}_\epsilon), \qquad \mathbf{Q} := (\mathbf{I} - \mathbf{F}^\top)^{-1}, \tag{6}$$

where $\mathbf{F}$ is the coefficient matrix to be optimized, in which elements without edges in the DAG are masked to zero. $\mathcal{N}(\boldsymbol{\mu}_\epsilon, \boldsymbol{\Sigma}_\epsilon)$ is the Gaussian distributions of the noise variables to be optimized. Then, the mean and covariance of observed variables $\mathbf{o}$ are

$$\boldsymbol{\mu}_\mathbf{o} = \mathbf{S}\mathbf{Q}\boldsymbol{\mu}_\epsilon =: \mathbf{A}\boldsymbol{\mu}_\epsilon, \qquad \boldsymbol{\Sigma}_\mathbf{o} = \mathbf{S}\mathbf{Q}\boldsymbol{\Sigma}_\epsilon\mathbf{Q}^\top\mathbf{S}^\top. \tag{7}$$

Given $N$ i.i.d. samples, let $\bar{\mathbf{x}}^\mathbf{o}$ be the sample mean and $\hat{\boldsymbol{\Sigma}}_\mathbf{o} := \frac{1}{N}\sum_{n=1}^{N}\left(\mathbf{x}_n^\mathbf{o} - \bar{\mathbf{x}}^\mathbf{o}\right)\left(\mathbf{x}_n^\mathbf{o} - \bar{\mathbf{x}}^\mathbf{o}\right)^\top$ be the centered sample covariance. The negative log-likelihood is

$$\text{NLL} = \frac{N}{2}\left[\log\det\boldsymbol{\Sigma}_\mathbf{o} + \text{tr}\!\left(\boldsymbol{\Sigma}_\mathbf{o}^{-1}\hat{\boldsymbol{\Sigma}}_\mathbf{o}\right) + (\bar{\mathbf{x}}^\mathbf{o} - \mathbf{A}\boldsymbol{\mu}_\epsilon)^\top\boldsymbol{\Sigma}_\mathbf{o}^{-1}(\bar{\mathbf{x}}^\mathbf{o} - \mathbf{A}\boldsymbol{\mu}_\epsilon)\right]. \tag{8}$$

## 4.3    STEP 5: REDUCE DYNAMIC CAUSAL GRAPH

The above steps identify all the GCMs from the DynaGCM. In this step, we reduce these GCMs to a dynamic graph, which includes two parts: (i) reduce GCMs to build the structure of DynaGCM, and (ii) traverse to search for all conditions for each dynamic edge. Roughly speaking, for all order pairs among the data variables, we add an edge between them if this edge exists in some GCMs. Further, if this edge does not exist in some GCMs, we label it as a dynamic edge. Finally, for each dynamic edge, we traverse all possible combinations of variables from small to large, and merge all valid combinations to build a logical expression as the condition of the dynamic edge. Although, per Definition 4, these conditions can in principle be inferred solely from changes in the

noise variable distributions, in this step we also require changes in the corresponding observed data distributions. Because the optimization in Equation 8 is nonconvex, the estimated noise distributions may be imperfect, potentially leading to erroneous conditional expressions. We provide the detailed algorithm in Appendix D and the theory analysis for the dynamic edges in Appendix F.

## 5 EXPERIMENTS

### 5.1 SYNTHETIC DATA

We begin with the synthetic data settings. Following Dong et al. (2023), we consider three types of causal graphs: (i) latent tree structures, (ii) latent measurement structures, and (iii) general latent structures. The tree models require each variable to have one parent; the measurement models allow each latent variable to have two pure children; and the general models further relax this rule. Examples are given in Appendix C. For each type, we consider 10 different DynaGCMs and dataset sizes $\{5k, 10k, 20k\}$. More details are provided in Appendix G.

Our method is general, allowing dynamic edges and flexible connections between latent and observed variables. Since existing methods do not address dynamic causal graphs with condition-labeled edges, there are no direct baselines, which also highlights both the novelty and validity of our framework. Therefore, we evaluate our approach in structure learning mainly by comparison with the ground truth. For the condition learning, to demonstrate the effectiveness of our solution, we propose an ablation method solely to infer the conditions from the changes of noise variable distributions in step 5, named **Our-S**.

**Structure learning/identification**   We use the *F1 score* to evaluate the performance of our method on the structure learning. In detail, F1 is the harmonic mean of precision and recall, i.e., $F1 = \frac{2 \times \text{Precision} \times \text{Recall}}{\text{Precision} + \text{Recall}}$. Precision measures the fraction of correctly identified edges among the edges retrieved by the algorithm. Recall measures the fraction of true edges that were successfully identified. The performances on structure learning are provided in Table 1. The high F1 scores across all settings show the effectiveness of our method. Our method achieves slightly lower scores on general models than both tree models and measurement models, demonstrating the robustness of our method on complicated scenarios. Among different sizes of data, our method still achieves comparable performance, which means our method can identify the structure even under a small data size.

**Condition learning**   We use the *Accuracy* (Acc) to evaluate the performance on condition learning. Following Dong et al. (2025), we rely on 30 random starts and choose the noise variable distributions with the best likelihood, because of the nonconvexness of Equation 8. The results on the condition learning are provided in Table 2. It shows that Acc for condition learning and F1 for structure learning follow the same trend, implying that error is dominated by structure learning rather than parameter learning in our method. Although the Acc on tree models > measurement models > general models, our method remains robust, effectively learning the conditioning variables of dynamic edges under complex settings. Moreover, our method achieves better Acc compared to Our-S, showing the effectiveness of considering changes on observed data variable distributions in Step 5.

Table 1: F1 on the structure learning.

| Data Settings | | Our |
|---|---|---|
| *Latent+tree* | 5k | **0.96** (0.03) |
| | 10k | **0.97** (0.01) |
| | 20k | **0.99** (0.00) |
| *Latent+measm* | 5k | **0.88** (0.09) |
| | 10k | **0.91** (0.05) |
| | 20k | **0.95** (0.04) |
| *Latent+general* | 5k | **0.79** (0.12) |
| | 10k | **0.83** (0.10) |
| | 20k | **0.91** (0.05) |

Table 2: Acc on the condition learning.

| Data Settings | | Our | Our-S |
|---|---|---|---|
| *Latent+tree* | 5k | **0.74** (0.15) | **0.66** (0.12) |
| | 10k | **0.79** (0.10) | **0.75** (0.13) |
| | 20k | **0.85** (0.09) | **0.78** (0.10) |
| *Latent+measm* | 5k | **0.66** (0.13) | **0.52** (0.16) |
| | 10k | **0.70** (0.12) | **0.65** (0.14) |
| | 20k | **0.73** (0.07) | **0.68** (0.11) |
| *Latent+general* | 5k | **0.51** (0.19) | **0.40** (0.18) |
| | 10k | **0.53** (0.13) | **0.44** (0.17) |
| | 20k | **0.57** (0.15) | **0.49** (0.11) |

## 5.2 REAL-WORLD DATA

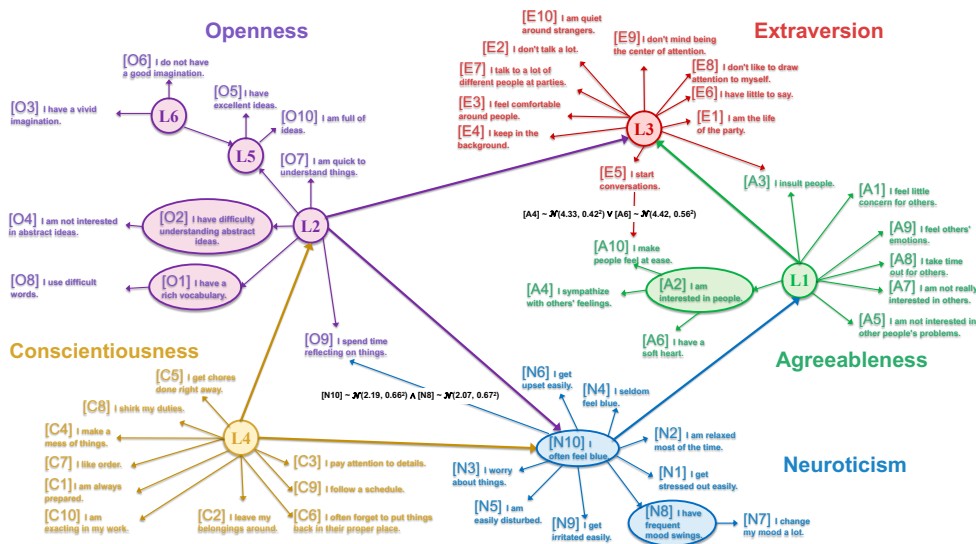

Figure 4: Estimated dynamic causal graph on real-world data.

In personality psychology research, the Big Five personality trait model, with the acronym OCEAN (Openness, Conscientiousness, Extraversion, Agreeableness, and Neuroticism), is the most common scientific model for measuring and describing human personality traits. To measure these traits, the factor analysis technique (Wright, 2017) is employed, which treats these five traits as latent variables that cause their factors, and thereby utilizes factor markers as a proxy to reconstruct the five traits (Goldberg, 1992). Another perspective is network theory (Cramer et al., 2012), which proposes that the five traits emerge from the causal structure of these factor markers. Note that Cramer et al. (2012) argues that these causal networks are dynamic among individuals, which also aligns with the dynamic causal graph of our methodology.

In this work, we adopt the Big Five dataset https://openpsychometrics.org/_rawdata/, which contains 19719 samples on 50 factor variables. Each personality trait has 10 factors with values in $\{1, 2, 3, 4, 5\}$, indicating the degree of the trait. Following Dong et al. (2023; 2025), we preprocess the data to unit variance and use GIN (Xie et al., 2020) to determine the remaining directions of the output MEC.

The results are shown in Figure 4. The 50 factors mainly group into 5 sub-groups, consistent with factor analysis (Goldberg, 1992). Moreover, factors for Openness, Neuroticism, and Agreeableness form networks, in line with network theory (Cramer et al., 2012). We also identify two dynamic edges in the results: one linking Neuroticism and Openness, and another linking Extraversion and Agreeableness. These edges show that reflection is more likely under higher neurotic states, and that conversations become more comforting when individuals show empathy. Detailed conditions and discussions are provided in Appendix G.2.

## 6 CONCLUSION & LIMITATION & FUTURE WORKS

In this work, we proposed a dynamic causal discovery method for identifying a linear latent Gaussian dynamic causal model (DynaGCM). Our method provides a novel paradigm for modeling changing mechanisms and supports decision-making in complex real-world scenarios. Specifically, we generalized the classic directed acyclic graph into a dynamic causal graph and employed observed distributions as conditions, allowing dynamic edges to be explicitly activated or deactivated. We further established identifiability results to ensure the correct recovery of the dynamic causal graph. Empirical results on the Big Five dataset demonstrated the effectiveness of our approach. Limitation and future works are discussed in Appendix H.

## ETHICS STATEMENT

This work investigates causal discovery using a publicly available Big Five personality questionnaire dataset (public URL provided in the paper). The data are fully anonymized and were originally collected with notice that responses may be used for research. We do not collect any new data or interact with participants, and we make no attempt at re-identification. Analyses are reported at aggregate levels and we avoid any claims that could stigmatize individuals or groups. According to our institution's policy, secondary analysis of publicly available anonymized data is typically exempt from human-subjects review; documentation will be provided upon request. We comply with the dataset's terms of use and release only code and derived artifacts necessary for reproducibility, without redistributing raw personal data. This research complies with the ICLR Code of Ethics, and all authors have read and acknowledged the Code.

## REPRODUCIBILITY STATEMENT

We have taken extensive steps to ensure the reproducibility of our work. Our dynamic causal discovery algorithm is detailed in Sections 4, and experiments are provided in Section 5. We further detail the hyperparameters, settings, datasets, and implementation details in Appendix G. Randomness in all experiments is controlled through the setting of seeds. The real-world experiment is based on a publicly available Big Five dataset, where the available link is provided in Section 5. To further support reproducibility, we provide anonymous source code in https://anonymous.4open.science/r/dynamiccausaldiscovery-7065/

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

# A   RELATED WORK

Our work aims to identify changing causal mechanisms with the dynamic causal graph, which relates to the following two fields: (i) causal discovery, (ii) parameter identification, as well as (iii) dynamic and condition-based causal discovery. First, causal discovery helps in discerning whether one variable influences another and thereby recovers the structure of the dynamic causal graph. Second, parameter identification helps in evaluating the extent of causal effects and states of variables, thereby helping in reducing the conditions in the dynamic causal graph. Last, we discuss current causal discovery methods in dynamic and condition-based settings.

**Causal Discovery**   Existing causal discovery methods (Zanga et al., 2022; Wang et al., 2024) mainly fall into three categories: constraint-based, score-based, and functional-based methods. (i) The constraint-based methods utilized tools, like the independence test and the rank deficiency, to determine the existence of latent variables and MEC, which usually adopt Meek rules (Meek, 1995) further to determine the directions of undirected edges in MEC. Typical independence test-based methods include PC (Spirtes et al., 2000), FCI (Spirtes et al., 1995), PCMCI (Runge et al., 2019b), tsFCI (Entner & Hoyer, 2010), and so on. The rank deficiency-based methods (Dong et al., 2023) use the rank test to replace the independence test, which provides more statistical information from observed data, and thereby these methods could identify causal graphs from more general environments. For instance, Huang et al. (2022) discovers latent hierarchical causal structure with the rank constraints. (ii) The score-based methods utilize score functions -such as BIC score (Schwarz, 1978), BDeu score (Heckerman et al., 1995), generalized score functions (Huang et al., 2018), and so on- to evaluate the score of a given structure and search for the best MEC. Typical methods include GES (Alonso-Barba et al., 2013), ExactSearch (Karan & Zola, 2016), and so on. (iii) The functional-based methods utilize an assumed function to describe the data generation process, where the causal structure is set as the parameters of this function. Building upon this, one can solve the function to learn the causal relationships. Typical methods include PNL (Zhang & Hyvärinen, 2009), LiNGAM (Shimizu et al., 2006), ANM (Peters et al., 2014), and so on.

Compared to score-based methods and functional-based methods, the constraint-based methods show excellent capacity in complicated real-world scenarios with latent variables. Though the functional-based methods also could discover causal structure with latent variables, they make strong assumptions about the data generation process, which is usually violated in reality. As a result of it, our method solves the dynamic causal discovery problem with the constraint-based method and the rank deficiency test.

**Parameter Identification**   To estimate the parameters and the noise variable distributions with the existence of latent variables, lots of methods are devised. For instance, one approach is to utilize the factor analysis technique, which describes a given set of observed variables by identifying latent variables and parameters. Typical methods includes Reiersøl (1950), Williams (2020), and so on. Furthermore, some approaches introduce the Overcomplete Independent Components Analysis (OICA) technique to solve this identification problem, such as Adams et al. (2021b), Salehkaleybar et al. (2020), Hoyer et al. (2008), and so on. In addition, there are also solutions to identify the parameters with causal inference techniques (Pearl, 2009a), which estimate the parameters as causal effects with do-calculus, instrumental variables, and so on. Typical methods include Tian & Pearl (2002), Jung et al. (2020), and so on. Conversely, some research first converts the DAG to its variants-such as Ancestral Graph (AG) (Richardson & Spirtes, 2002), Acyclic Directed Mixed Graph (ADMG) (Pearl, 2009b), and so on-then solves the identification problem with graph criteria like half-trek (Foygel et al., 2012; Barber et al., 2022), G-criterion (Brito & Pearl, 2002a), and so on. Typical methods includes Drton et al. (2011), Brito & Pearl (2002b), and so on. Since our dynamic causal discovery method requires identifying not only the parameters and noise variable distributions, but also the causal structure in general cases, we follow this approach.

**Dynamic and Condition-based Causal Discovery.**   A growing line of work studies causal discovery under *changing* mechanisms. CD-NOD (Zhang et al., 2017) and its follow-ups (Huang et al., 2020) leverage heterogeneity or nonstationarity across domains/time to recover skeletons and orientations by exploiting independent changes in causal modules, often using an observed or inferred domain index to summarize mechanism shifts. In partially observed settings, recent identifiability results give graphical conditions under non-Gaussian or heterogeneous errors, but do not provide

a unified representation of when an edge is *active* versus *inactive* across conditions (Adams et al., 2021a). Time-series approaches (e.g., PCMCI and variants) embed dynamics into lags to detect contemporaneous and lagged links; extensions to semi-stationary or periodic regimes inherit this temporal framing (Runge et al., 2019a; Runge, 2020). By contrast, we target a *single* dynamic causal graph whose edges carry explicit condition labels, aiming to capture compact, time-agnostic physical rules: mechanisms are switched by conditions rather than by time indices. Moreover, while many heterogeneous or multi-view methods assume known environment labels or a fixed number/structure of regimes, our framework *infers* both the assignments and the number of regimes from data and then reduces them to condition-labeled edges in one graph.

## B  ADDITIONAL INFORMATION

**Definition 9 (Minimal-Graph Operator (Huang et al., 2022))** *Given two atomic covers $\mathbb{A}, \mathbb{B}$ in $\mathcal{D}$, we can merge $\mathbb{A}$ to $\mathbb{B}$ if the following conditions hold: (i) $\mathbb{A}$ is the pure children of $\mathbb{B}$, (ii) all elements of $\mathbb{A}$ and $\mathbb{B}$ are latent and $|\mathbb{A}| = |\mathbb{B}|$, and (iii) the pure children of $\mathbb{A}$ form a single atomic cover, or the siblings of $\mathbb{A}$ form a single atomic cover. We denote such an operator as minimal-graph operator $\mathcal{O}_{min}(\mathcal{D})$.*

**Definition 10 (Skeleton Operator (Huang et al., 2022))** *Given an atomic covers $\mathbb{A}$ in a graph $\mathcal{D}$, for all $\mathbf{v}_i \in \mathbb{A}$, $\mathbf{v}_i$ is latent, and all $\mathbf{v}_j \in PCh(\mathbb{A})$, such that $\mathbf{v}_i$ and $\mathbf{v}_j$ are not adjacent in $\mathcal{D}$, we can draw an edge from $\mathbf{v}_i$ to $\mathbf{v}_j$. We denote such an operator as skeleton operator $\mathcal{O}_s(\mathcal{D})$.*

## C  GRAPH EXAMPLES

### C.1  GRAPH EXAMPLES FOR LATENT TREE GRAPH STRUCTURES

Please refer to Figure 5.

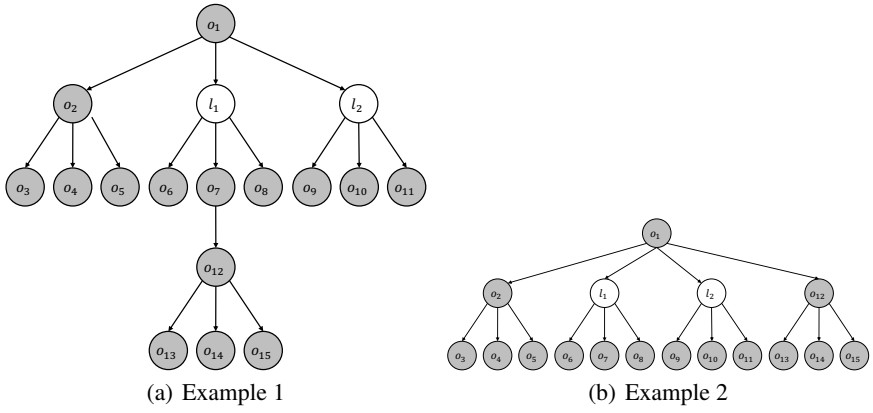

(a) Example 1                    (b) Example 2

Figure 5: Examples of Latent Tree Graph Structures

### C.2  GRAPH EXAMPLES FOR LATENT MEASUREMENT GRAPH STRUCTURES

Please refer to Figure 6.

### C.3  GRAPH EXAMPLES FOR GENERAL LATENT GRAPH STRUCTURES

Please refer to Figure 7.

## D  PSEUDOCODE OF STEP 5

Please refer to Algorithm 4.

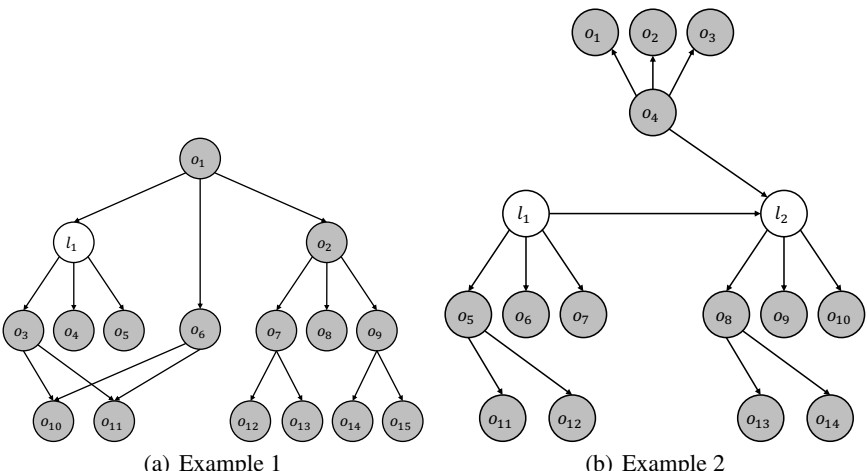

(a) Example 1          (b) Example 2

Figure 6: Examples of Latent Measurement Graph Structures

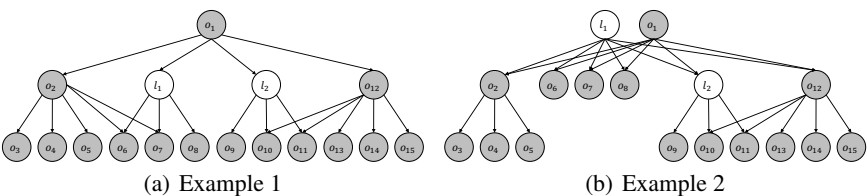

(a) Example 1          (b) Example 2

Figure 7: Examples of General Latent Graph Structures

# E  DETAILS ON IDENTIFICATION AND REDUCTION

**Detailed explanation of Figure 1.**  In the circuit example, the observed data x can only follow three distinct causal structures. Formally, we start from Equation 1, and substitute the condition-specific indicators $f_{ver}(\cdot)$. This leads to the following generation rules:

(i) If $v_{s1} = 1$, the edge $e_{s1 \to L}$ is active and Equation 3 holds. (ii) If $v_{s1} = 1 \wedge v_{s2} = 0$, then Equation 4 applies. (iii) If $v_{s1} = 0 \wedge v_{s2} = 0$, then Equation 5 applies.

Each observed sample x is generated under exactly one of the three conditions in the circuit environment: Condition 1 ($v_{s1} = 1$), Condition 2 ($v_{s1} = 1, v_{s2} = 0$), or Condition 3 ($v_{s1} = 0, v_{s2} = 0$). This mapping ensures that every instance can be uniquely aligned with one of the three DAGs. Thus, each assignment of conditions corresponds to a specific GCM.

**Connection to GCM discovery.**  Each of the three equations, together with its DAG, forms a linear Gaussian causal model (GCM). To recover these GCMs from data, we apply the RLCD algorithm (Dong et al., 2023), which identifies the Markov equivalence class from rank-based constraints, and then use the parameter estimation method in Dong et al. (2025) to recover coefficients and noise terms.

**Identification of assignments.**  In the main text, we state that different GCMs induce different Gaussian distributions over the observed variables $\mathbf{x^o}$. Here we provide more details. According to Dong et al. (2025), if a set of GCMs is identifiable, then their corresponding observed distributions must differ. This implies that data generated under different latent conditions can, in principle, be separated. In addition, Yakowitz & Spragins (1968) proves that mixtures of multivariable Gaussian distributions are identifiable (up to label permutation). Combining these results, the assignment of samples to GCMs is guaranteed to be identifiable under our assumptions. In practice, we adopt the Expectation-Maximization (EM) algorithm to estimate the parameters of the mixture model and to

---

**Algorithm 4:** REDUCEDYNAMICCAUSALGRAPH

---

**Require:** DAGs $\mathcal{D} = \{\mathcal{D}_{1:K}\}$ on variables $\mathbf{v}$, where the observed variables is $\mathbf{o}$ and the latent variables is $\mathbf{l}$, the observed variable distributions $\mathcal{N}_{\mathbf{o}}^{1:K}$ computing by the estimated edge–coefficient matrices $\{\mathbf{F}^k\}_{k=1}^K$ and noise variable distributions $\mathcal{N}_{\boldsymbol{\epsilon}}^{1:K}$

**Ensure:** dynamic causal graph $\mathcal{G}$

 1: create a dynamic causal graph $\mathcal{G}$ among $\mathbf{v}$ with empty edge
 2: **for all** $(\mathbf{v}_i, \mathbf{v}_j) \in \mathbf{v}, i \neq j$ **do**
 3:    **if** $\exists k \in \{1, \cdots, K\}, \exists \mathbf{e} \in \mathbf{e}^k : f_{\mathbb{S}^k}(\mathbf{e}) = i \wedge f_{\mathbb{T}^k}(\mathbf{e}) = j$ **then**
 4:      *// — add an edge to the dynamic causal graph*
 5:      add an edge $\mathbf{e}$, a map $\mathbf{e} \to i$, and a map $\mathbf{e} \to j$ into the $\mathbf{e}, \mathbb{S}, \mathbb{T}$ of $\mathcal{G}$, respectively
 6:      **if** $\exists k' \in \{1, \cdots, K\}, \nexists \mathbf{e} \in \mathbf{e}^k : f_{\mathbb{S}^k}(\mathbf{e}) = i \wedge f_{\mathbb{T}^k}(\mathbf{e}) = j$ **then**
 7:        *// — this edge is dynamic, create an empty condition to this edge*
 8:        add condition variable $\mathbf{c}^{\mathbf{e}}$ with empty condition into $\mathbb{C}^{\mathbf{e}}$, add maps $\mathbf{v}_j \to \mathbf{c}^{\mathbf{e}}$ into $\mathbb{L}^{\mathbf{e}}$
 9:        let $\mathcal{D}_{\mathbf{e}}$ be the DAG set that exists the edge $\mathbf{e}$
 10:        $s \leftarrow 1$
 11:        **while** $s \leq |\mathbf{o}| - 1$ **do**
 12:          **for all** $s$ observed variable distributions $\mathcal{N}_{\mathbf{o}}^k$ except the target variable $\mathcal{N}_{\mathbf{o}_j}^k$ from one of DAGs $\mathcal{D}$ that has the edge $\mathbf{e}$ **do**
 13:            **if** (i) in the remaining $\mathcal{D} \setminus \{\mathcal{D}^k\}$, both these $s$ observed variable distributions and their noise variable distributions hold, then these must be the edge $\mathbf{e}$ exists. (ii) in addition, if the edge $\mathbf{e}$ **not** exists, both these $s$ observed variable distributions and their noise variable distributions must **not** hold. (iii) no true subset of these $s$ observed variable distributions satisfies both the (i) and (ii) requirement. **then**
 14:              reduce these $s$ observed variable distributions in logical and relationship $\wedge$ equation, and put this equation into the condition $\mathbf{c}^{\mathbf{e}}$ with a logical or relationship $\vee$
 15:           **end if**
 16:         **end for**
 17:         $s \leftarrow s + 1$
 18:        **end while**
 19:      **end if**
 20:    **end if**
 21: **end for**
 22: **return** $\mathcal{G}$

---

recover the assignment of each sample. The Bayesian Information Criterion (BIC) can be further used to select the optimal number of mixture components.

**Reduction to a dynamic causal graph.** Once the assignments and individual GCMs are identified, the next step is to merge them into a unified dynamic causal graph. Inspired by Definition 4, we interpret the differences among the estimated GCMs as evidence for dynamic edges. Specifically, if the existence of an edge depends on a minimal set of distributions of noise variables, then the observed distributions of these variables define a condition $\mathbf{c}^{\mathbf{e}} \in \mathbb{C}^{\mathbf{e}}$. Formally, for each pair of variables, we traverse the estimated GCMs: if the edge appears in some GCMs but not in others, we label it as dynamic and record the corresponding conditions. By merging across all groups, we obtain the final dynamic causal graph $\mathcal{G}$ with condition-labeled edges.

## F  THEORY ANALYSIS FOR DYNAMIC EDGES OF DYNAMIC CAUSAL GRAPH

**Definition 11 (hard-intervention (Pearl, 2009b))** *For the DynaGCM in Definition 2, a hard intervention on $\mathbf{v}_k$ that sets it to $\mathbf{x}_k$, denoted $\mathrm{do}(\mathbf{v}_k = \mathbf{x}_k)$, constructs the intervened model by replacing the structural equation of $\mathbf{v}_k$ with the constant $\mathbf{v}_k := \mathbf{x}_k$, i.e., removing all incoming arrows into $\mathbf{v}_k$; all other equations for other variables and the noise variable distributions remain unchanged.*

**Definition 12 (Non-condition)** *Given $\mathbf{x}^{\mathbf{o}}$, generated in the environment with changing mechanisms, existed a dynamic edge $\mathbf{e}_{i \to j}$ with a condition $\mathbf{c}^{\mathbf{e}_{i \to j}}$, which is according to Equation 1. For all the*

*original sample values $x_1^o, \ldots, x_j^o, \cdots, x_{|\mathbf{v}|}^o \in \mathbf{x^o}$ obeyed the $c^{e_{i \to j}}$, let $\hat{x}_j^o$ represent the differ sample values corresponding to $x_j^o$. Similarly, let $x_1^{o\prime}, \ldots, x_j^{o\prime}, \cdots, x_{|\mathbf{v}|}^o{}' \in \mathbf{x^o}$, $\hat{x}_j^{o\prime}$, and $x_i^{o\prime}$ be the values with respect to the case not obeyed the $c^{e_{j \to i}}$.*

*We can determine the non-condition of $c^{e_{i \to j}}$ with respect to dynamic edge $e_{i \to j}$, if the following condition holds: (i) for all $x_1^o, \ldots, x_j^o, \cdots, x_{|\mathbf{v}|}^o$, hard-intervent $x_j^o$ to $\hat{x}_j^o$ in the environment, then the value of $v_i$ is still the same with $x_i^o$; (ii) for all $x_1^{o\prime}, \ldots, x_j^{o\prime}, \cdots, x_{|\mathbf{v}|}^o{}'$, hard-intervent $x_j^{o\prime}$ to $\hat{x}_j^{o\prime}$ in the environment, then the value of $v_i$ sometimes be identical or different with $x_i^{o\prime}$.*

*That is, (i) the sample values of $v_i$ are invariant to $v_j$ with hard-intervention in Equation 1 under $c^{e_{i \to j}}$; (ii) the sample values of $v_i$ sometimes be identical or different with hard-intervention in Equation 1 when $c^{e_{i \to j}}$ is not met.*

**Proposition 1 (Identificaiton on Condition of DynaGCM)** *Given the dynamic causal graph $\mathcal{G} = (\mathbf{v}, \mathbf{e}, \mathbb{C}^\mathbf{e})$, which has a dynamic edge $e_{i \to j}$, assuming that data generation process adheres to Equation 1, we can deduce that $c^{e_{i \to j}} \notin \mathbb{C}^\mathbf{e}$ if and only if: across all $\mathbf{x^o} \in \mathbf{x^o}$, when hard-intervent to $v_j$ with Equation 1, (i) the sample values of $v_i$ are always invariant with $f_{ver}(\mathbf{x^o}, c^{e_{j \to i}}) = 1$ and (ii) the samples value of $v_i$ are not always invarient with $f_{ver}(\mathbf{x^o}, c^{e_{j \to i}}) = 0$.*

**Proof:** Let $x_1^o, \ldots, x_j^o, \cdots, x_{|\mathbf{v}|}^o$ and $\hat{x}_j^o$ represent the original sample values and hard-intervent sample values under condition $c^{e_{i \to j}}$ in Definition 12, respectively. Let $x_i^o$ and $\hat{x}_i^o$ be the sample values on $v_i^o$ of Equation 1 with the above original and intervention values, respectively. Similarly, for the case not obeyed $c^{e_{j \to i}}$, let $x_1^{o\prime}, \ldots, x_j^{o\prime}, \cdots, x_{|\mathbf{v}|}^o{}'$ and $\hat{x}_j^{o\prime}$ represent the original sample values and hard-intervent sample values, respectively. Let $x_i^{o\prime}$ and $\hat{x}_i^{o\prime}$ be the values of $v_i$ .

$\Longrightarrow$: If $c^{e_{i \to j}} \notin \mathbb{C}^\mathbf{e}$, then according to Definition 12 and Equation 1, it necessitates that (i) $x_i^o = \hat{x}_i^o$ and (ii) $x_i^{o\prime}$ sometimes be identical or different. Conversely, according to Equation 1, if (i) the sample values of $v_i$ are sometimes changing with $f_{ver}(\mathbf{x^o}, c^{e_{j \to i}}) = 1$, then the the causal effect from $v_j^o$ may influence the sample value of $v_i^o$ under Equation 1 in sometimes, (i.e., $x_i^o \neq \hat{x}_i^o$), leading to a contradiction; (ii) the sample values of $v_i$ are always invariant with $f_{ver}(\mathbf{x^o}, c^{e_{j \to i}}) = 0$, then the the causal effect from $v_j^{o\prime}$ will always not influence the sample values of $v_i^o$ under Equation 1, (i.e., $x_i^{o\prime} = \hat{x}_i^{o\prime}$), leading to a contradiction. Hence, if $c^{e_{i \to j}} \notin \mathbb{C}^\mathbf{e}$, it logically follows that there must be: (i) the sample values of $v_i$ are always invariant with $f_{ver}(\mathbf{x^o}, c^{e_{j \to i}}) = 1$ and (ii) the sample values of $v_i$ are not always invarient with $f_{ver}(\mathbf{x^o}, c^{e_{j \to i}}) = 0$.

$\Longleftarrow$: If (i) the sample values of $v_i$ are always invariant with $f_{ver}(\mathbf{x^o}, c^{e_{j \to i}}) = 1$, then $x_i^o$ and $\hat{x}_i^o$ will yield identical values, as corresponding to Equation 1, (ii) the sample values of $v_i$ are not always invarient with $f_{ver}(\mathbf{x^o}, c^{e_{j \to i}}) = 0$, then $x_i^{o\prime}$ sometimes be identical or different with $\hat{x}_i^{o\prime}$. This implies considering $c^{e_{i \to j}}$ in Equation 1, we still can not determine the mechanism that generates $v_i$, allowing us to infer $c^{e_{i \to j}} \notin \mathbb{C}^\mathbf{e}$. $\qquad\square$

This proposition inspires a methodology to identify the conditions of a dynamic edge of the dynamic causal graph from the data generation by DynaGCMs with Equation 1, by traversing all combinations of data variables and discerning whether this combination is non-condition with the identified parameters and noise variable distributions from the previous steps of our algorithm. We provide the pseudocode in Algorithm 4.

# G  EXPERIMENT DETAILS

## G.1  SYNTHETIC DATASET

The average generated parameters of Equation 1 are set to: 15 observed variables and 2 latent variables; 2 GCMs and each of them has 1 different dynamic edge; 2 conditions for a dynamic edge; $[-2.5, 2.5]$ causal strength for each edge; Gaussian distribution with $[-100, 100]$ mean and $[1, 5]$ variance for each noise variable. We use 10 random seeds to generate the data for each setting and report the mean performance as well as standard deviation.

**Structure Learning.** In the experiments on synthetic data verifying the structure learning, to align the latent variables in the output graph of GCMs with the latent variables in the ground truth graph, similar to Dong et al. (2023), combinations and permutations of latent variables are considered. In detail, we first pad each result by adding latents that have no edge to any other variables to match the number of latents in the ground truth graph. Then, if the number of latents is more than that of the ground truth, all different combinations will be tried. Finally, we try all the different permutations of latent variables to test the F1 score.

**Condition learning.** In condition learning experiments, if the relative error between the parameter values of two distributions does not exceed $10\%$, we consider them to be equal, as it is difficult for a floating-point number obtained through optimization to perfectly match the desired value.

## G.2    DETAILS OF DYNAMIC EDGES IN REAL-WORLD DATA

The first dynamic edge is between "[N10] I often feel blue." and "[O9] I spend time reflecting on things." with condition $[N_{10}] \sim \mathcal{N}(2.19, 0.66^2) \;\wedge\; [N_8] \sim \mathcal{N}(2.07, 0.67^2)$. Since 2.19 and 2.07 are relatively high values in the dataset, this implies that individuals reflect more when in a higher state of neuroticism.

The second dynamic edge is between "[E5] I start conversations." and "[A10] I make people feel at ease." with condition $[A_4] \sim \mathcal{N}(4.33, 0.42^2) \;\vee\; [A_6] \sim \mathcal{N}(4.42, 0.56^2)$. This indicates that when individuals care about or empathize with others, the conversations they initiate are more comforting, which aligns with common expectations.

# H    LIMITATION AND FUTURE WORKS

Our current study has several limitations. First, the theoretical results are developed under linear Gaussian assumptions, which may not fully capture more complex causal mechanisms in practice. Second, the algorithm may face higher computational cost when the number of variables and conditions increases. Third, the evaluation is still limited in scope, and further tests on broader real-world datasets are needed.

In future work, we aim to relax the current assumptions and develop more flexible and powerful solutions. In particular, we are interested in applying our framework to challenging settings such as the cold-start problem in world model construction for reinforcement learning, where dynamic causal discovery could provide a principled way to bootstrap model-based agents in open-ended environments.

# USE OF LLMS

We employed large language models (LLMs) solely to polish the writing of this paper, such as improving grammar, clarity, and readability. The models were not used for generating original idea-sexperiments, analyses, or results. All scientific contributions, methods, and conclusions presented in the paper are entirely the work of the authors.

