# OpenReview forum: "Causal Discovery under Changing Mechanisms: A Unified Graphical Approach"
_ICLR.cc/2026/Conference — Submitted to ICLR 2026_

### Official Review · Reviewer_s5fK · 2025-10-28

**Soundness:** 2
**Presentation:** 1
**Contribution:** 2
**Rating:** 2
**Confidence:** 4

**Summary:**

The authors propose a causal discovery method for dynamical causal graphs (DynaGCM), which are graphs with contextually activated edges controlled via the state of some (latent) variables. Specifically the authors consider linear Gaussian dynamic graphs and employ an EM algorithm to iteratively label and identify a set of latent graph models from a set of given data points. For this, the authors heavily rely on the prior RLCD algorithm of Dong et al. (2025) and theoretical results of Dong et al. (2023) and Yakowitz & Spragins (1968) to establish graph identifiability. An additional graph reduction step is added, which build a DynaGCM from the individually identified Gaussian causal models.



Experiments are conducted over simple synthetic and real-world data, showing good F1 scores for structure identification and moderate accuracy for condition learning on the synthetic data. For the real-world Big Five dataset plausible predictions (as not ground-truth exists) are obtained, identifying two reasonable conditional link.

**Strengths:**

The paper tackles the important scenario of discovering dynamical causal graphs, while most approaches in causal discovery (CD) assume target static graph structures, which is however often not reflected in real-world data.

The presented initial example is intuitive and clearly conveys the need for considering contextual conditioning during CD.

The high-level steps of the proposed EM iteration algorithm are clearly laid out. Within the algorithm, the authors seem to soundly leverage identification results and methods of prior works.

The obtained results seem to show a well working of the method for identifying the overall causal structure and the switching conditions. Experiments on the real-world big five dataset seem to furthermore discover a reasonable graph structure while identifying two plausible conditional links from the data.

**Weaknesses:**

While the authors tackle the generally important problem of CD under dynamic causal graph, the paper is generally vague in its presentation of assumption and formalization of the target model. The heavy use of existing methods and formal results in section 4 is not matched by the formal rigor of prior sections and assumptions used throughout the proofs, making the paper feel unpolished in its current state and the soundness of results hard to judge. In the following I list several inconsistencies, which might all be fixable on their own. However, the sheer number of suggestions makes it impossible for me to assess the overall soundness of the proposed formalism and method.



Minor point: I believe the 'on' and 'off' labels for condition 2 in figure 1 should be inverted to align with the below graphic and description in the text.



**Definition 1.** The definition of dynamic causal graphs is confusing to me, as it merges a formal definition with actual values in the third bullet point. Generally, I was unable to understand what the domain of $\mathbb{C}^e$ should be, considering this first definition. Is it a set of sets of active edges/structural equations? Is it defining conditions under which an edge becomes active? How do the logical and- and or-operators evaluate given a Gaussian distribution? (The discussion in section 3 seems to indicate that related to variable instantiation). I would generally like to advice separating the formal definition and the example in figure 2 and, moreover, define what the 'and' and 'or' operators in the then following example are meant to express. Similarly, the example suddenly switches from a discrete scenario to a continuous variable domain. The authors might want to describe the transition more clearly.



Minor point: The authors mention a placeholder value of '9999' in figure 2 to indicate an open link. I would like to suggest to formally assign a special symbol instead of a value in $\mathbb{R}$ to indicate open links and to avoid 'magic numbers'.



**Definition 2.** Next, the assignment of variable values in $v\_i$ in definition 2 seems to be based on a set of observed data. This somehow inverses the model->data generating process, as now the model 'is generated from data'. While this might be valid approach for a causal discovery (CD) approach, it skips the step of formally defining a assumed underlying *true* model, whose parameters are usually only regressed afterwards from data.



The result of $f\_{ver}$  is dependent on "if the observed variables in sample $x^o$ satisfy c...". Since, $x^o$ was previously defined as a set of observations, it is unclear to me whether the condition is all-quantified or whether a single example fulfilling the condition suffices (which I assume to be the case given the context). I would like to suggest to explicitly mention this in the text.

Next, I wonder what the $f\_{ver}$ in the subscript of the sum in equation 1 iterates over. Is it the $x^o$ or c's? Given that the proposed DynaGCM should resemble a standard causal graph under a given latent context, I am under the expression that for every edge ij only a single weight $\alpha\_{ij}$ might be assigned under any particular context. Considering the definitions up until definition 2 and since $v_i$ in eq. 1 seems not to be conditioned on any particular instantiation of $l$, it is unclear to me how the particular weights are selected. Generally, I can conditions assumed to be mutually exclusive (or otherwise consistent to each other).



Minor point: Edge weights in general additive noise models are usually assumed to take arbitrary values in $\R$. I wonder whether the restriction of weights $\alpha$ to [0, 1] in definition 2 could simply be lifted.



**Definition 4.** For inferring definitions the authors mention the "multivariable Gaussian distributions [to] hold". I am wondering is meant by Gaussian distribution to 'hold'. Does it assume that a variable can only exhibit a single distribution? What is the variable's distribution given that some of its parent edges are inactive?



**Related Work.** While I find it irritating that related work is not discussed within the main text, slightly pushing on page limitations, the authors sparely cite contextual CD, apart from CD-NOD. In particular recent work of [1] considers with contextual causal discovery under a non-timeseries setting and from joint data, citing further prior work, e.g. in the subfield of LDAG discovery.

Similarly, the scenario presented in figure 1 and mentions of root-cause attributions is similar to the of the formalism of 'meta-causal models' [2] where switching variables are assumed to assert effects directly on the edge E->L. In classical SCM this might be represented via contextual dependencies as done in this work. While the authors argue that only actual effects should be represented in the resulting graph, there might be an argument on whether the contextual conditioning variables (e.g. the switches) should always be connected to the effected L, as they always assert influence by deciding whether or not the other factors become connected to the particular variable.



**Section 3.** The condition of eqs 3 and 4 are not mutually exclusive (both consider $v\_{s1}=1$). Similarly the case $v\_{s1}=0 \land v\_{s2}=1$ is not covered by the equations.



**Section 4.** The writing in section 4 contains multiple typos and is not self-contained.

* l. 239: "Pure children (P Ch) _are_ defined"

* Def. 7: "Let v in v denote[s] the"

* "top()" in definition 8 is undefined.

The direction step in section 4.2 seems to imply that edges between any two variables $x_i, x_j$ over all graph instantiations in an GCM are assumed to always be directed the same and never switch in direction. The authors might want to make this assumption explicit in the beginning.

To my understanding the authors argue that the joint set of the different discovered PAGs can be used to determine the direction of all edges. It is not clear to me why this should always the case, e.g. in cases where an edges can not be defined in any of the graphs.



In section 4.3 the authors mention that their approach "traverse[s] all possible combinations of variables from small to large". Could the authors elaborate which metric is used for this determining this ordering and which assumptions come with it. Could this step interact with var-sortability conditions [3]?



Other typos:

* Proposition 1: "Identificaiton" -> "Identification"
* Definition 12: 'the differ' -> 'a different'



[1] Günther, W., Popescu, O.I., Rabel, M., Ninad, U., Gerhardus, A. and Runge, J., 2024. Causal discovery with endogenous context variables. *Advances in Neural Information Processing Systems*, *37*, pp.36243-36284.

[2] Willig, M., Tobiasch, T., Busch, F.P., Seng, J., Dhami, D.S. and Kersting, K., Systems with Switching Causal Relations: A Meta-Causal Perspective. In *The Thirteenth International Conference on Learning Representations*.

[3] Reisach, A., Seiler, C. and Weichwald, S., 2021. Beware of the simulated dag! causal discovery benchmarks may be easy to game. *Advances in Neural Information Processing Systems*, *34*, pp.27772-27784.

**Questions:**

I do not have so many general questions, as I would rather like to ask the authors to clarify the many remarked unclarities mentioned in the weaknesses section above. More concretely, answering the following questions might help understanding of the method:

1) Could the authors clarify the definition of $\mathbb{C}^e$ and role of $f\_{ver}$  with regard to determining edge weights?
2) Could the authors elaborate on relation to the mentioned related work and how their approach differs in assumptions with regard to the required data and/or modeling assumptions?
3) Could the authors elaborate how leveraging multiple PAGs in Sec. 4.2 guarantees full identification of all edge directions? Why can't it be, that some edges might not be identifiable even across multiple DAGs?

---

### Official Review · Reviewer_Kttp · 2025-11-01

**Soundness:** 4
**Presentation:** 4
**Contribution:** 3
**Rating:** 8
**Confidence:** 4

**Summary:**

The papers presents an interesting formalization on dynamic causal graphs taking into account different triggers for edge activations under the assumption of linear latent Gaussian relationships.

**Strengths:**

The paper shows clear problem statement and definitions with well-motivated example on electrical circuit. It shows clear theoretical results with algorithmic pipeline and performed both synthetic experiments and real-world data on personality traits.

**Weaknesses:**

**Scope of “dynamic causal models.”** The topic is compelling but broad in the sense depending on the “condition,” both the **edge set**—and possibly even the **variable set**—could change. In the limit, such flexibility risks a hypothesis class that can explain almost any dataset, which weakens falsifiability and identifiability. Given the paper contributes to a framework to study dynamical causal model, I would encourage the authors delineate the intended scope (what can vary, what is fixed) and articulate the inductive biases that keep the problem learnable. In particular, it would also be good to clarify how dynamic causal model differs from a mixture of Bayesian model, and showcase practical advantages.

**Notations.**  In Eqs. (3–5), the triggering conditions differ but the parent set is written uniformly as $\mathrm{PA}(v_i)$. Although the authors noted that $\mathrm{PA}(v_i)$ depends on the triggering condition, the current notation obscures this and can be misleading. I encourage the authors to adopt a notation that makes the dependence explicit, so it is clear that different conditions induce different parent sets. A good mathematical notation for dynamic causal models should make it immediately visible that different triggering conditions $\Rightarrow$ different parents/mechanisms.

**Beyond linear–Gaussian / perfect identifiability in Step 1.** The linear–Gaussian assumptions and the “perfect identifiability” guarantee for condition assignment are theoretically clean, but in practice dynamic systems are noisy and finite-sample. I encourage the authors, in future work, to **relax linear–Gaussianity** and replace hard identifiability with **quantified uncertainty** over condition classes—e.g., probabilistic assignments with calibrated uncertainties—and to **propagate this uncertainty** through subsequent structure learning. While perfect identifiability is appealing, **well-calibrated finite-sample uncertainty** can be more useful empirically for dynamic causal modeling.

**Questions:**

See above.

---

### Official Review · Reviewer_fkbV · 2025-11-02

**Soundness:** 3
**Presentation:** 1
**Contribution:** 2
**Rating:** 0
**Confidence:** 4

**Summary:**

The paper discusses situations where existence of the edges in a causal graph seems to depend on external circumstances. Authors leverage existing results in latent structure discovery and mixture of gaussians discovery to devise a causal graph discovery algorithm for this setting.

**Strengths:**

There seems to be a coherent story in regards to the previous work, although presentation could be improved to show this more clearly. The experiments are carefully designed and executed, with the code available. I do not have a generally positive evaluation of the paper at the moment, but I am open to reconsideration if there is a serious misunderstanding on my part, which I find unlikely given my reasonable familiarity with the field.

**Weaknesses:**

There are two major issues with the paper that deems it not suitable for publication at the moment.

1. The problem setting seems to be in fundamental conflict with the existing formulations of causal models.
2. The paper is notoriously hard to follow for someone familiar with the field, let alone a layman.

### 1. Issues with the problem setting:
The motivating example can be found in Pearl (2009) under 10.1.2 "Preemption and the Role of Structural Information". It is meant to demonstrate that even with perfect knowledge of a causal model it is nuanced to decide if a certain variable is "cause" since not in all realizations it would serve as a but-for cause for the studied effect.

The authors, use this opportunity to motivate "dynamic causal graphs", a graphical model with labeled directed edges. The labels are conditions which determine whether the edge exists. These conditions are on the distribution of a set of "observable variables" $X^O$, e.g., $A \to B$ exists if $C$ follows $\mathcal{N}(0,1)$. The formulation allows for more complex conditions, e.g., composition of conditions with logical operators, or considering joint distributions.

At first glance, the motivating example seems to suggest that "causation" depends on realization, e.g., whether a certain switch is responsible for the light being on depends on the state/realization of the other switch. This reading motivates Pearl's counterfactual-centered formalization of causality, e.g., the role of quantities such as probability of necessity (PN) and probability of sufficiency (PS) in deciding credit/blame attribution. However, the authors reading is fundamentally different, inspiring to define "dynamic causal graphs" such that existence of the edges depends on the probability distribution of other variables, suggesting that somehow edges/causal dependence is informed by statistic of other variables. Definition 2 makes this confusion more explicit:

**Definition 2 (Linear Latent Gaussian Dynamic Causal Model (DynaGCM))**

Suppose a dynamic causal graph $ G = (\mathbf{v}, \mathbf{e}, \mathbf{C^e}) $, each data variable is generated by the equation:

$$
v_i = \sum_{v_j \in Pa(v_i), f_{ver}(\mathbf{x^o}, \mathbf{c}^{e_{j \to i}})=1}
\alpha^{e_{j \to i}} v_j + \epsilon_i
$$

where $Pa(v_i)$ are all variables that are possible parents of $V_i$, and $f_{ver}(\mathbf{x^o}, \mathbf{c}^{e_{j \to i}})$ is a binary indicator such that $f_{ver}(\mathbf{x^o}, \mathbf{c}^{e_{j \to i}}) = 1$ if the observed variables in sample $\mathbf{x^o}$ satisfy $\mathbf{c}^{e_{j \to i}}$ and $f_{ver}(\mathbf{x^o}, \mathbf{c}^{e_{j \to i}}) = 0$ otherwise. Moreover, each noise variable \( \epsilon \) is a Gaussian-distributed variable.

In this definition, somehow $f_{ver}(\mathbf{x^o}, \mathbf{c}^{e_{j \to i}})$ is supposed to determine whether a *sample* is drawn from a certain distribution. In fact, it is mathematically problematic once you consider $f_{ver}$ as a mapping that takes points in the domain of $X^O$ and evaluates the conditions for the edges, because the sample itself has no information about the distribution it is drawn from. Arguably, this definition is not constructive as it offers no concrete way of generating samples from such model. This makes me seriously concerned about everything else in the paper.

### 2. Paper's presentation
The paper does not make the content accessible to readers that are even slightly unfamiliar with the previous work and context. The motivating examples must be a way to ground intuition about definitions and motivate the theoretical results, however, the motivating example here (the circuit) is quite distant from the linear gaussian models, so it does not really serve the purpose, and it can even confuse the readers. Much of the content of the theory is adapted from previous work on gaussian model identification, and it is not possible to evaluate the novelty of the work, even if the serious issues about the fundamentals were not in the picture. Concretely, about novelty I would like to ask if is a accurate reading that this work:
a. The data is a mixture of gaussian distributions, so the first stage discovers the mixture components using existing classical source separation methods, and the second stage relies primarily on the existing work to determine the structure based on the single gaussian distribution.
b. The graph aggregation is the novel component here, deciding what are the conditions for each edge. Arguably, this step does not add any value beyond summarization of the graphs in the first step. The Dynamic Causal Graphs are motivated, defined, and discovered in this paper, although the motivation and definition are seriously questionable.

**Questions:**

Please address concerns above.

---

### Official Review · Reviewer_r6GY · 2025-11-10

**Soundness:** 1
**Presentation:** 1
**Contribution:** 2
**Rating:** 2
**Confidence:** 3

**Summary:**

The paper proposes a dynamic causal graph model intended to represent causal mechanisms that change across latent conditions, and presents an algorithm to discover such graphs by clustering samples (via Gaussian mixture identifiability) and then performing causal discovery within each cluster. The authors claim to provide theoretical identifiability conditions and demonstrate empirical results on synthetic data and a personality questionnaire dataset. The dynamic graph is described as a generalization of DAGs with condition labels on edges. The experiments primarily evaluate recovery accuracy on simulated latent-variable structures and provide one illustrative real-world application.

**Strengths:**

- The paper provides a complete end-to-end algorithmic pipeline (Assignment → Structure Identification → Parameter Estimation → Graph Reduction), rather than presenting only theoretical concepts. The workflow is clearly structured, and the flowchart in Figure 3 makes the procedural steps understandable.

**Weaknesses:**

1.	Very limited novelty relative to existing literature.
The core idea—cluster samples by mixture models and run causal discovery per cluster—is already standard in the heterogeneous causal discovery literature. For example, existing works on nonstationary or heterogeneous mechanisms provide similar conceptual frameworks using environment or regime assignments. The proposed dynamic causal graph appears to be primarily a notational rephrasing and does not provide new identifiable structure beyond prior mixture-of-GCM approaches. The conceptual contribution is therefore minimal.
2.	Overstated theoretical claims.
The identifiability claims rely on strong assumptions (linear Gaussian latent models, special rank and trek conditions) that are already known in prior work. The paper claims “general” identifiability, but the theoretical scope is highly restricted. No comparison is made to known identifiability frameworks such as nonstationary ANMs or latent GCM identifiability criteria.
3.	Experimental evaluation is very limited.
All synthetic experiments use extremely constrained settings with predefined known GCM families. There is no comparison to baselines that are designed for heterogeneous or condition-dependent causal discovery (e.g., CD-NOD, causal representation methods, time-varying causal discovery frameworks). Without such comparisons, the claim of practical usefulness is unsupported.
4.	Ungrounded claims without empirical or theoretical justification.
The manuscript repeatedly asserts that the proposed model “unifies” causal mechanisms and can “handle real-world dynamics,” yet the only real-world demonstration is a personality questionnaire dataset, which lacks ground-truth causal structure and is interpreted post-hoc. This does not validate the method.
5.	The central graph representation is not justified.
The proposed dynamic causal graph assumes that condition-labeled edges represent meaningful mechanism changes. However, the paper does not justify why a single shared graph with switching edges is more appropriate than simply learning distinct graphs per regime. In fact, the presence of non-unique edge activation across samples indicates that no single causal structure applies universally, making the graph abstraction questionable. The “merging” procedure appears arbitrary and does not solve the underlying ambiguity in structure.

Weaknesses (Minor)
- Abstract does not report any concrete numerical results.
- The purpose of introducing a unified dynamic graph is unclear; the paper never demonstrates scenarios where this representation is necessary over simpler alternatives.
- The interpretation of estimated dynamic edges in the real-world dataset is speculative and not validated.
- Many statements in the introduction (e.g., about generality or expressiveness) are not supported by theory or experiments.

**Questions:**

1. What is the actual novelty compared to existing heterogeneous / nonstationary causal discovery frameworks? Please detail what is new beyond clustering + causal discovery + edge comparison.

2. Why should one prefer a single dynamic causal graph abstraction instead of simply maintaining multiple discovered causal graphs, which seems more faithful to the data?

3. How do you justify the interpretation of real-world dynamic edges when no ground truth is available?

---

### Meta-Review · Area_Chair_Lz83 · 2026-01-02

**Summary:**

This paper proposes a dynamic causal graph model that captures causal mechanisms varying across latent conditions. It introduces an algorithm for learning such graphs by first clustering samples and then performing causal discovery within each cluster. The work leverages identifiability results for Gaussian mixtures and claims theoretical identifiability guarantees. The work presents empirical evaluations on a synthetic dataset and a real-world personality questionnaire dataset. The proposed dynamic graph is formulated as a generalization of directed acyclic graphs (DAGs), where edges are annotated with condition-specific labels.

The paper received 4 reviews with 1 recommending accept and 3 recommending rejection. The major points of contention were as follows:

* Limited novelty of the work. Several reviewers pointed out that the major results of the paper in light of the known assumptions are relatively well known. I agree with the assessment and while I do believe that the work is moving towards the right direction, the presentation and the writing leaves a lot to be desired.

* Lack of baselines and enough experimental evidence even in the current experiments.

* Several reviewers also point out the the heavy use of notations and formal results. As reviewer s5fK puts it and I quote " these are not matched by the formal rigor of prior sections and assumptions used throughout the proofs, making the paper feel unpolished in its current state and the soundness of results hard to judge." and I got the same feeling during my reading.

Ov erall, I feel that the paper is a bit undercooked and thus does not meet the bar for acceptance in its current form.

**Reviewer Concerns:**

There was no rebuttal provided so this section is not applicable.

**Reviewer Scores:**

There was no rebuttal and thus no discussion.

---

### Decision · Program_Chairs · 2026-01-26

Reject